# Regeneration Process of an Autologous Tissue-Engineered Trachea (aTET) in a Rat Patch Tracheoplasty Model

**DOI:** 10.3390/bioengineering11030243

**Published:** 2024-02-29

**Authors:** Shun Iwasaki, Koichi Deguchi, Ryosuke Iwai, Yasuhide Nakayama, Hiroomi Okuyama

**Affiliations:** 1Department of Pediatric Surgery, Osaka University Graduate School of Medicine, 2-2 Yamadaoka, Suita 565-0871, Japan; iwasaki@pedsurg.med.osaka-u.ac.jp (S.I.); deguchi@pedsurg.med.osaka-u.ac.jp (K.D.); 2Research Institute of Technology, Okayama University of Science, 1-1, Ridaicho, Kita-Ku, Okayama 700-0005, Japan; iwai@ous.ac.jp; 3Osaka Laboratory, Biotube Co., Ltd., 3-10-1 Senriyama-Higashi, Suita 565-0842, Japan; y.nakayama@biotube.co.jp

**Keywords:** tissue engineering, tracheal regeneration, tracheal transplantation, patch tracheoplasty, congenital tracheal stenosis, rat model, regeneration process

## Abstract

The treatment of long-tracheal lesion is difficult because there are currently no viable grafts for tracheal replacement. To solve this problem, we have developed an autologous Tissue-Engineered Trachea (aTET), which is made up of collagenous tissues and cartilage-like structures derived from rat chondrocytes. This graft induced successful long-term survival in a small-animal experiment in our previous study. In this study, we investigated the regeneration process of an aTET to attain reproducible success. We prepared an aTET by using a specially designed mold and performed patch tracheoplasty with an aTET. We assigned twenty-seven rats to three groups according to the three types of patch grafts used: aTET patches (the aTET group), fresh tracheal autograft patches (the Ag group), or polylactic acid and polycaprolactone copolymer sheets (the PPc group). In each group, gross and histological evaluations were performed at 1 month (*n* = 3), 3 months (*n* = 3), and 6 months (*n* = 3) after implantation. We obtained high survival rates in all groups, but only the PPc group attained thick tracheal walls with granular tissues and no tracheal regeneration. On the other hand, the aTET and Ag groups reproducibly achieved complete tracheal regeneration in 6 months. So, an aTET could be a promising candidate for tracheal regeneration grafts.

## 1. Introduction

A radical therapy for long-segment tracheal lesions caused by congenital tracheal stenosis, tumors, trauma, and so on remains challenging. “Long segment” generally means more than one-half of the total tracheal length in adults and more than one-third in children [1,2,3,4,5,6]. If the lesion is not long-segment, tracheoplasty, end-to-end anastomosis, or slide tracheoplasty are currently clinically performed. In contrast, in long-segment cases, we have not developed a remedy for this lesion and need to find a way of interposing luminal structures to cure it completely.

Various grafts have been proposed in both tracheoplasty and circumferential tracheal transplantation. As synthetic prostheses or scaffolds, many materials, namely, stainless steel [7,8]; steel coil [9]; glass [7]; polyethylene [10]; silicone [11,12]; Teflon [13]; Marlex [14]; and Dacron [15], have been used. These materials often have been used together with autogenous free-flap tissues of the costal cartilage [16]; tracheal wall [17]; fascia [18]; small intestine submucosa [2,19]; pericardium [20]; palatal mucoperiosteal [21]; dura mater [22]; bladder mucosa [23]; aorta [24,25,26]; and esophagus [27,28]. On the other hand, the rapid development of tissue-engineering technology has recently provided new methods for circumferential tracheal replacement [29]. Archna et al. [30] provided an overview of tracheal transplantation with a tissue-engineered trachea. Although many grafts have been suggested, very few of them demonstrated long-term survival. In particular, only two reports on long-term survival in small-animal models have been published [31,32], including our previous study, which is the only study on tissue-engineered graft use [32]. To develop a new tracheal graft with reproducible long-term survival, there are three problems that must be addressed: (1) creating a graft with sufficient strength, (2) minimizing foreign body reactions, and (3) promoting the regeneration of the tracheal structures [2,3,4,5,6,33,34,35,36,37,38,39,40,41,42,43,44,45,46,47].

To solve these problems, we developed an original graft of an autologous Tissue-Engineered Trachea (aTET) in a previous study. This graft has a tubular structure composed of autologous collagenous tissue and cartilage-like constituents fabricated from rat chondrocytes. When using this aTET for circumferential tracheal implantation in a rat model, it survived for 8 months, with no lethal tracheal collapse or granulation and with the regeneration of tracheal epithelium and the maintenance of cartilaginous tissue [32]. But approximately 60% of the models died within one postoperative day due to tracheal obstruction [32], and we have not yet determined the regeneration process of the aTET in detail because of the low survival rates of the models. So, we conducted this study to clarify the regeneration process of the aTET by performing patch tracheoplasty, for which the survival rate is much higher compared to the circumferential tracheal implantation model.

## 2. Materials and Methods

To clarify the process of tracheal regeneration in various grafts implanted in vivo, we performed patch tracheoplasty on rats using the following three grafts: (1) aTET patches (aTET group), (2) fresh tracheal autograft patches (Ag group), and (3) polylactic acid and polycaprolactone copolymer sheets (PPc group). Rats were euthanized at 1 month, 3 months, and 6 months postoperatively via intraperitoneal administration of an overdose of pentobarbital (100 mg/kg) or carbon dioxide gas asphyxiation. The regeneration processes of the three types of grafts were compared through gross and histological evaluation of the grafts.

### 2.1. Graft Preparation

As previously reported [32], we manufactured an original tracheal mimetic tissue of an aTET containing a ring-shaped cartilage-like structure and collagenous tissue with enough strength to maintain the tracheal structure. Briefly, the process of manufacturing the aTET involved three steps. First, we made cartilage rings consisting of chondrocytes from the costal cartilage of Lewis rats. Second, three cartilage rings were combined with a rod covered with a silicon cylinder. Finally, the construct was embedded into the dorsal subcutaneous pouch of a Lewis rat for 6 to 8 weeks. By harvesting the construct, we could obtain aTET by removing a rod from it. In addition, we used an autograft resected from the defect of the recipient (the Ag group) and polylactic acid and polycaprolactone copolymer (PPc group), respectively.

### 2.2. Tracheoplasty Procedure

We employed male Lewis rats (LEW/CrlCrlj, 8–10 weeks, 250–400 g, Charls River Laboratories, Wilmington, MA, USA) as recipients. The rats were anesthetized with 4% isoflurane inhalation and subcutaneous injections of atropine (0.05 mg/kg) and leptin (0.015 mg/kg) as premedication. To maintain anesthesia, we used intraperitoneal injections of medetomidine (0.075 mg/kg), midazolam (1 mg/kg), and butorphanol tartrate (1.25 mg/kg). We maintained a body temperature of 38 degrees throughout the surgery. The surgical site was shaved, and a suprasternal midline incision was created through the skin and underlying tissues in a sterile fashion. The fascia of the sternohyoid muscle was also incised in the midline. The sternohyoid muscles were split to expose the ventral side of the trachea. We bluntly separated the adventitia from the trachea, and a full-thickness defect (2 × 4-5 mm (3 cartilage rings)) was created in the ventral tracheal wall one or two cartilages below the thyroid gland. Autograft and the aTET were cut into quarters, and we fixed one of them in the defect with 8-0 Nylon interrupted suture. PPc was implanted directly into the rat trachea, not into the dorsal pouch before patch tracheoplasty. If there was oozing or bleeding, we performed astriction. Before closing the defect, we removed the intraluminal sputum as much as possible because asphyxia via intraluminal sputum was the main cause of surgical death. If respiration effort worsened or dyspnea was seen after closing the defect, we cut the suture immediately and opened the defect of the trachea to remove the intraluminal sputum. After the respiratory status was confirmed to be stable for a few minutes, the incision was closed. The sternohyoid muscles and the skin were sutured using interrupted 4-0 Nylon. Throughout this surgery, great care was taken not to damage the connective tissue around the trachea by grasping or rubbing it. Figure 1 shows an overview of our experimental design.

After the operation, 5 mL of 5% glucose solution was injected subcutaneously. We maintained a body temperature of 38 degrees until the rats woke up. We gave them feed that was ground and moistened only within 24 h after the operation, and from then on, we gave them normal feed. Carprofen (5.0 mg/kg/day) was used as an analgesic for three days postoperatively, and we used erythromycin (20 mg/kg/day) as an antibiotic and atropine (0.1 mg/kg/day) for the suppression of secretion for seven days postoperatively. These three drugs were injected subcutaneously. In addition, we gave rats inhalers supplied with 0.25 mL of 0.05% Salbutamol Sulfate, 1 mL of 0.02% Bromhexine Hydrochloride, and Budesonide (0.1 mg/kg/day, two times per day, respectively) for seven days to make it easier to expectorate and to suppress the thickening of tracheal edema. At 1 month, 3 months, and 6 months after the operation, the rats were euthanized using intraperitoneal injections of pentobarbital sodium (50–100 mg/kg) or carbon dioxide gas, and we explanted the grafts (*n* = 3 in each period).

### 2.3. Evaluations of the Explanted Graft

We observed the intraluminal and outer-surface findings of the explanted grafts, serving as gross findings. The explanted grafts were fixed in 4% paraformaldehyde for twelve to twenty-four hours and preserved in 70% ethanol. We performed paraffin embedding of them, and they were sectioned in 6 to 8 μm thick serial sections because of failure to cut the specimens with cartilage tissues into the usual thickness of 4–5 μm.

Histological evaluations were performed using Hematoxylin and Eosin staining for general findings and Alcian blue staining for goblet cells and cartilage substrates. Masson’s Trichrome staining was used for fibrous tissue, and Von Kossa staining was used for calcification. Von Kossa staining was not performed for the PPc group because we could not detect cartilage-like constructions in the graft. In addition, we performed immunofluorescence staining using anti-alpha Tubulin antibody (1:50, ab7291, Abcam, Cambridge, UK,) for ciliated cells, anti-CD68 (1:50, ab125212, abcam) for M1 macrophages, and anti-alpha SMA antibody (1:100, 904601, BioLegend, San Diego, CA, USA) for myofibroblast cells as primary antibodies and using rabbit anti-mouse IgG H&L (Alexa Fluor^®^ 594) (1:1000, ab150128, Abcam) for anti-alpha Tubulin, goat anti-rabbit IgG H&L (Alexa Fluor^®^ 488) (1:500, ab150077, Abcam) for anti-CD68 antibody, and donkey anti-mouse IgG H&L (Alexa Fluor^®^ 594) (1:500, ab150108, Abcam) for anti-alpha SMA antibody as secondary antibodies. Regarding myofibroblast cells, when granulation tissue had formed, the fibroblasts differentiated and myofibroblasts appeared, which were observed and evaluated qualitatively. As evaluations of the degree of acute inflammation severity, we calculated submucosal thickness and the number of infiltration cells (all cells were counted except for those with spindle-shaped nuclei in fibrous tissue and erythrocytes) and vessels at the center of the grafts in each group using Hematoxylin and Eosin staining. We used a fluorescence microscope (Eclipse Ts2-FL, Nikon^®^, Minato City, Japan), analysis software (NIS-Elements, version 5.20.00, Nikon^®^), and ImageJ software (version 1.53k). Note that a statistical evaluation of the calculated values was not performed because the sample size was too small for statistical evaluation.

### 2.4. Ethics

All animal experiments were performed under general anesthesia in accordance with the Guide for the Care and Use of Laboratory Animals, published by the United States National Institutes of Health (NIH Publication No. 85-23, revised 1996), and the ARRIVE guidelines. The research protocol was approved by the Ethics Committee of Osaka University (No: 02-044-002).

## 3. Results

### 3.1. Survival Rate

The overall postoperative survival rate was 100% (25/25). Two rats died intraoperatively, both due to asphyxia from sputum in the trachea. One rat belonged to the Ag group, and the other belonged to the aTET group.

### 3.2. Gross Appearance of the Graft

Gross findings concerning the internal and external surfaces of the harvested grafts in each group one month after surgery are shown in Figure 2. We obtained successful engraftments in all survival cases. And no instances of leakage, graft displacement, tracheal collapse, or granulation that could lead to luminal narrowing were observed on either the outer side or intraluminal surfaces. In each group, all grafts exhibited no deformation or collapse of the trachea that would compromise respiratory status, and adhesion between each graft and recipient tissue did not affect the maintenance of the lumen. We saw neither atrophy nor deformation of the cartilage in the Ag and aTET groups. In the PPc group, we could not detect a cartilage-like structure in the PPc graft during the observation period.

### 3.3. Histological Examinations

< Hematoxylin and Eosin and Masson’s Trichrome staining >

The Hematoxylin and Eosin and Masson’s Trichrome staining results are shown in Figure 3A. The Masson’s Trichrome staining results show the area surrounded with a square in the Hematoxylin and Eosin staining image. In models where the epithelium could not be included in the same image of Masson’s Tricrome staining due to the thickness of the submucosal tissue, an image showing the epithelium was added ((l), (q), and (r) in Figure 3A). In the Ag and aTET groups, the tracheal epithelium was completely regenerated at 6 months. But in the PPc group, the tracheal epithelium was not regenerated during the observation period. In addition to this, the submucosal layer was thickened 3 months after the surgery, and Masson’s Trichrome staining was positive in the thick tissue, especially around PPc or its degradation products. Figure 3B shows the thickness measurements of the submucosal layer in the graft area in each group. The measurements were taken during the Hematoxylin and Eosin staining procedure. In all the models, the submucosal tissue thickness was thinnest at one month postoperatively. The PPc group had a more intense increasing tendency than the Ag and aTET groups. Figure 4B,C show the trends of the number of inflammatory cells and blood vessels. The aTET group had a decreasing trend, while the autograft group had an increasing trend in inflammatory cells and a decreasing trend in blood vessels. No clear trend was detected.

< Immunofluorescence staining for the CD68 and alpha SMA antibodies >

Figure 4A shows the results of immunofluorescence staining for the anti-CD68 antibody and anti-alpha SMA antibody. In the Ag and aTET groups, only one or two CD68-positive cells were seen under a high-power field, and the number of alpha SMA-positive cells increased gradually after one month postoperatively without CD68-positive cells. In the PPc group, in the inner-side area, which was separate from the PPc or its degradation products due to thickening of the submucosal tissue, CD68- and αSMA-positive cells remained in low numbers during the observation period. In contrast, on the outer side where PPc was present, the number of positive cells was still negligible at 1 month, but after 3 months, when PPc degradation products began to be seen, both positive cells were confirmed. CD68-positive cells were observed evenly in the outer area, but most of the αSMA-positive cells were mainly located at slight distance from the PPc degradation products.

< Alcian Blue staining and Immunofluorescence staining for the alpha Tubulin >

Alcian Blue staining and Immunofluorescence staining images for alpha-Tubulin are shown in Figure 5. Because the PPc showed thickening of the submucosal tissue, two images are presented for the 3- and 6-month models: one for the inner area and the other for the outer area ((n) and (o) in Figure 5). Immunofluorescence staining shows the area surrounded with a square in the Alcian Blue staining image. Alcian Blue staining remained positive in the cartilage tissue in the Ag and aTET groups till 6 months postoperatively, while no similar findings were observed in the PPc group throughout the observation period. In addition, the Alcian Blue staining in the Ag and aTET groups shows that some positive cells in the epithelium had increased in number at 6 months postoperatively. But in the PPc group, no positive cells were seen. The immunofluorescence staining for alpha-Tubulin showed that some positive cells were observed in the Ag and aTET groups 3 months postoperatively. And till 6 months, the luminal surface of the graft was completely occupied with positive cells in both groups. In contrast, no positive cells were identified in the PPc group even at 6 months postoperatively.

< Von Kossa staining >

Figure 6 shows the results of Von Kossa staining for the Ag and aTET groups. The recipient’s trachea was already positive for cartilage at the time of transplantation, and the graft cartilage remained positive for 6 months postoperatively in the Ag group. In the aTET group, calcification was first observed at 6 months (Figure 6f). No positive area in Von Kossa staining was confirmed, except for the cartilage.

< Summary of histological findings in each period >

All the groups exhibited no tracheal regeneration till one month. At three months, the epithelium partially regenerated and cartilage kept its structure in the aTET and Ag groups. The PPc group exhibited no tracheal regeneration and a trend of increasing submucosal thickness. Finally, at 6 months, the aTET and Ag groups achieved complete epithelium regeneration and retained cartilage, though inflammation and fibrosis were generally not seen. In the PPc group, no tracheal regeneration remained, and the submucosal layer was thickened via fibrosis, mainly around PPc.

**Figure 3 bioengineering-11-00243-f003:**
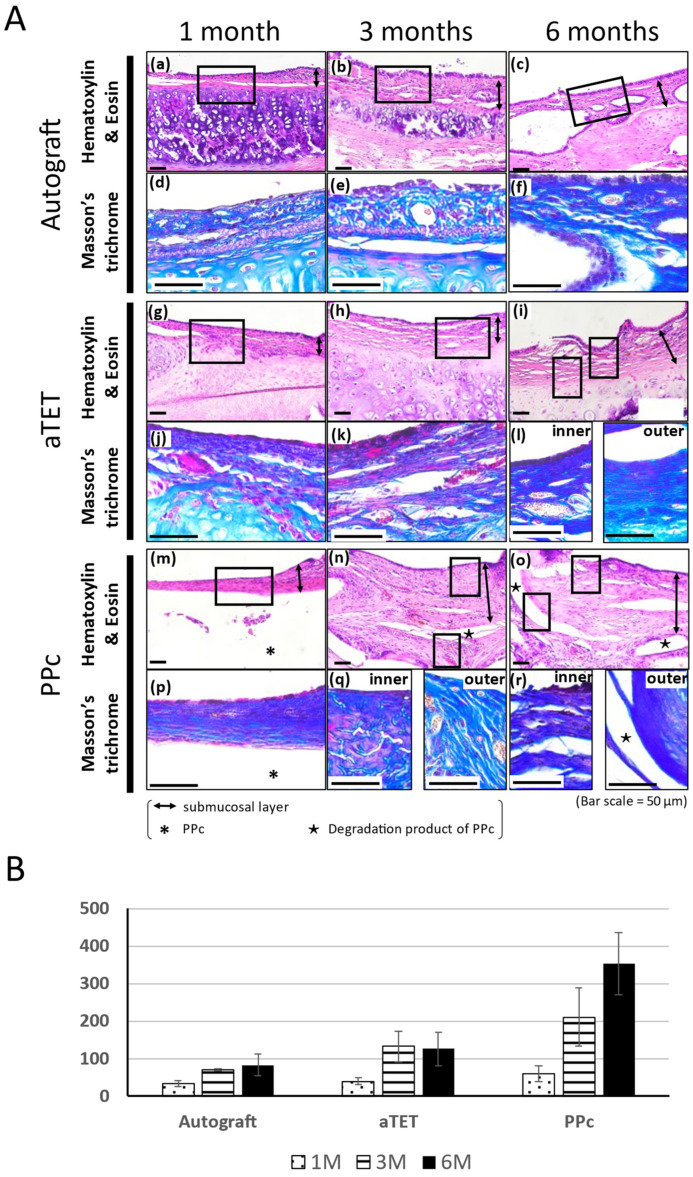
Histological analysis of tracheal wall and quantitative analysis of submucosa thickness changes. (**A**): Histological analysis using Hematoxylin and Eosin and Masson’s Trichrome staining. Images of the autograft group are shown in (**a**–**f**), images of the aTET group are shown in (**g**–**l**), and images of the PPc group are shown in (**m**–**r**). Hematoxylin and Eosin staining images are shown in (**a**–**c**), (**g**–**i**), and (**m**–**o**). Masson’s Trichrome staining images surrounded portions with squares are shown in (**d**–**f**), (**j**–**l**), and (**p**–**r**). One-month PO (postoperative) images are shown in (**a**,**d**,**g**,**j**,**m**,**p**). Three-month PO images are shown in (**b**,**e**,**h**,**k**,**n**,**q**). Six-month PO images are shown in (**c**,**f**,**i**,**l**,**o**,**r**). (**B**): Submucosal thickness at center of grafts.

**Figure 4 bioengineering-11-00243-f004:**
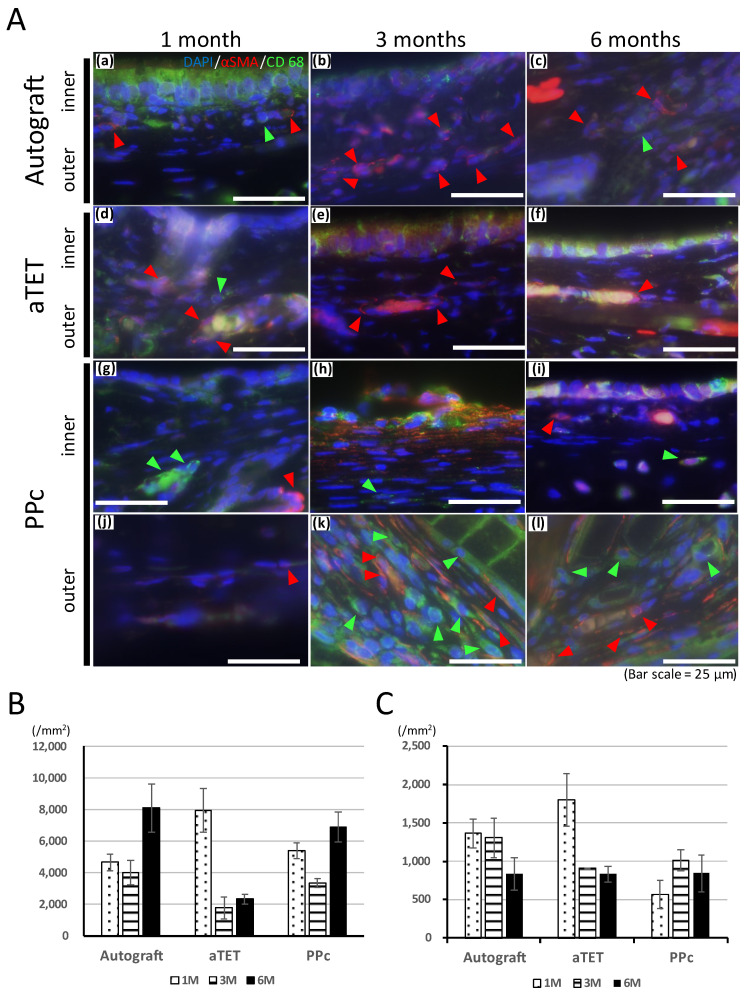
Qualitative analysis of inflammation. (**A**): Representative immunofluorescence images for CD68 and alpha SMA. Images of the autograft group are shown in (**a**–**c**), images of the aTET group are shown in (**d**–**f**), and images of the PPc group are shown in (**g**–**l**) (outer area; (**g**–**i**), inner area; (**j**–**l**)). One-month PO images are shown in (**a**,**d**,**g**,**l**). Three-month PO images are shown in (**b**,**e**,**h**,**k**). Six-month PO images are shown in (**c**,**f**,**i**,**l**). (red arrow head: alpha SMA, green arrow head: CD68). (**B**): Infiltration cells per 1 mm^2^ at center of grafts after Hematoxylin and Eosin staining. (**C**): Number of vessels at center of grafts.

**Figure 5 bioengineering-11-00243-f005:**
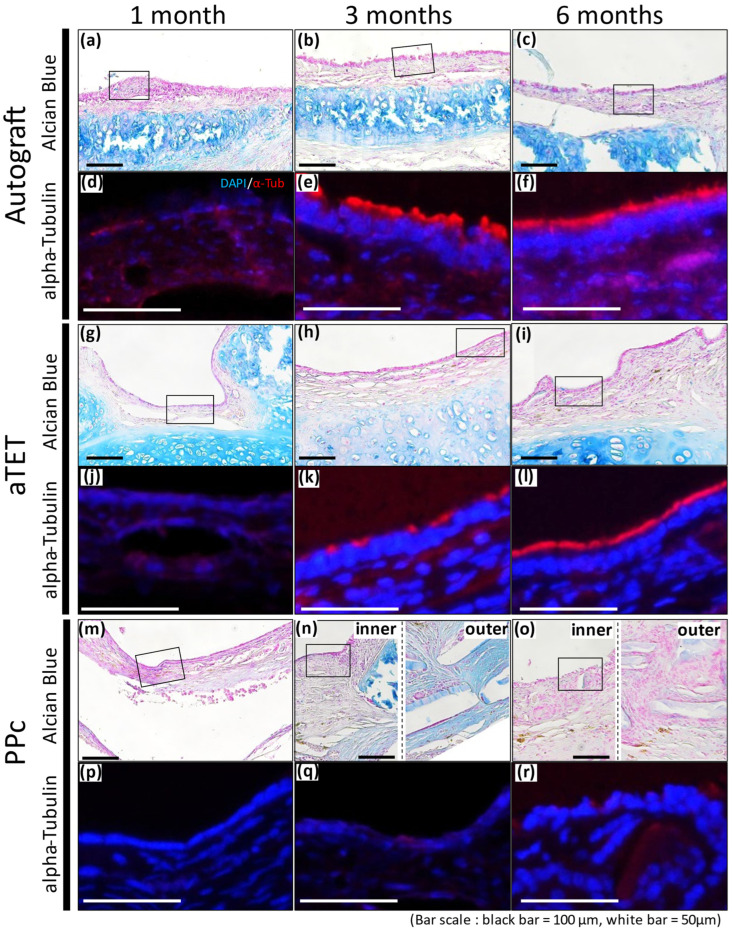
Alcian blue staining and representative immunofluorescence images for alpha-Tubulin. Autograft group images are shown in (**a**–**f**), aTET images are shown in (**g**–**l**), and PPc images are shown in (**m**–**r**). Alcian blue staining images are shown in (**a**–**c**,**g**–**i**,**m**–**o**). Immunofluorescence images for alpha-Tubulin staining images surrounded by portions with squares are shown in (**d**–**f**,**j**–**l**,**p**–**r**). One-month PO images are shown in (**a**,**d**,**g**,**j**,**m**,**p**). Three-month PO images are shown in (**b**,**e**,**h**,**k**,**n**,**q**). Six-month PO images are given in (**c**,**f**,**i**,**l**,**o**,**r**).

**Figure 6 bioengineering-11-00243-f006:**
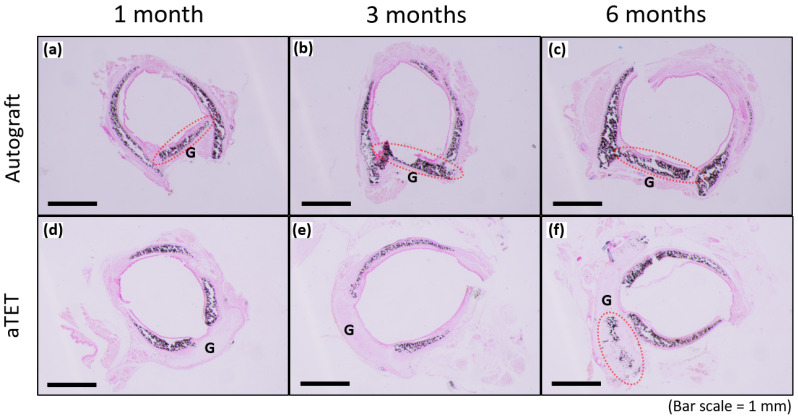
Von-Kossa staining. The aTET group images are shown in (**a**–**c**), and the autograft group images are presented in (**d**–**f**). One-month PO images are shown in (**a**,**d**), three-month PO images are displayed in (**b**,**e**), and six-month PO are presented in (**c**,**f**). (G: graft; red circles: calcification on grafts).

## 4. Discussion

Animal experiments with medium-sized or large animals such as dogs or rabbits are conducted frequently to develop suitable grafts for tracheal transplantation for long-segment lesions. However, few studies about the regeneration process of the implanted tracheal grafts have been reported. The low survival rate of circumferential tracheal replacements is one of the reasons for this death. Therefore, our first step should be to conduct experiments on small animals to study grafts that provide stable and high survival rates after surgery. At first, it is necessary to show that a graft can maintain strength as a trachea to withstand changes in airway pressure and minimize the tracheal wall thickening associated with inflammation due to foreign body reaction. Then, mucosal changes can be investigated grossly with bronchoscopy or computed tomography and histologically. V. Gretchen and P.R. Delaere et al. claimed that data from animal studies, including on what happened inside the grafts, were sparse and that the efficacy and safety of tracheal replacement with bioengineered tracheal grafts should be demonstrated [48,49]. We need to make obtaining reproducibility in animal experiments the highest priority before aiming at clinical application. In addition, the series of histological graft changes remains unclear and must constitute crucial knowledge because the trachea has unique features consisting of the environmental differences between the intraluminal surface exposed to inhaled air and the outer surface that comes into contact with living tissue [48].

Our previous study demonstrated that the preimplantation strength of an aTET was comparable with that of a rat trachea and that no disruption of tracheal structure occurred during the observation period [32]. In the present study, high survival rates were achieved in all groups regardless of the observation period, and all the surviving rats also showed no disruption of the tracheal structure or narrowing of the lumen after tracheoplasty (Figure 2). These results indicate that an aTET as well as an Ag and PPc may provide enough strength as a tracheal graft to withstand the change in intraluminal pressure, which may minimize the risk of asphyxia through the obstruction of sputum following the narrowing of the intraluminal space.

Foreign body reactions are an unavoidable problem when synthetic materials are used as scaffolds to withstand airway pressure. In this study, despite thickened tissue, no narrowing was seen because of thickening in the outside direction, the reason for which could not be clarified. If circumferential tracheal replacement was performed instead of patch implantation, the risk of intraluminal narrowing would be higher if using PPc instead of an aTEF because Figure 3B and Figure 4B show the former had a proinflammatory tendency, which the latter did not. In addition to this, as shown in Figure 4A, the area around PPc, unlike the inner side, showed a prolonged inflammatory tendency. This area is almost consistent with the area of thickened submucosal tissue in the PPc image in Figure 3. From these processes, it can be inferred that using artificial materials like the PPc clinically adopted in the airway environment leads to prolonged thickening of the airway wall and exposure to a high risk of intraluminal obstruction. Moreover, we speculated that after the PPc was encapsulated by granulation tissue, its degradation products caused the prolonged or new foreign body reactions. As shown in Figure 4C, the changes in the number of vessels are almost consistent with or do not show the trends of a proinflammatory tendency in each group because the Ag and aTET groups had a declining tendency and the PPc group had an increasing tendency concerning the number of vessels. Considering these results, in the Ag and aTET groups, the proinflammatory tendency gradually reduced; in the PPc group, the proinflammatory tendency was maintained throughout the study period. If we used the tracheal graft including artificial materials and it exhibited a persistent inflammatory response, we would have to consider the potential for airway obstruction due to granulation. Some reports have suggested that the uselessness of some immunosuppressants that may be needed when using artificial materials was one of the ideal features of a tracheal graft. Considering this background, it is desirable to develop grafts that do not incorporate artificial materials as scaffolds.

Finally, tracheal epithelium regeneration and cartilage retention have also been major challenges. But, as shown in Figure 3A and Figure 5, we could confirm that both problems were almost solved, which is the key to maintaining the intraluminal space. Tracheal epithelial regeneration contributes to preventing airway obstruction by sputum by maintaining the functions of sputum ejection performed by ciliary cells and of mucus production by goblet cells. The maintenance of cartilage contributes to the strength to withstand changes in airway internal pressure. Thus, an aTET has the potential to contribute to intraluminal patency both functionally and morphologically with complete tracheal regeneration. In the PPc group, such findings could not be confirmed. Many studies where stent insertion maintained the intraluminal space after tracheal transplantation have been reported, but stent insertion might be unnecessary if the cartilage tissue can be kept viable for a long time. Since intratracheal stenting does not fundamentally solve the problems that arise after implantation, and since stenting is not possible in small-animal experiments, we think that stenting should be used only as a temporary measure.

In this study, we presented how long an aTET graft takes to induce complete tracheal regeneration. We found out that the regeneration time in the aTET group was similar to that in the Ag group. The importance of intra-graft vessels regarding the resumption of blood supply has been reported [50], and an aTET has blood vessels because this graft is developed in vivo. The presence of these vessels may have contributed to the acquisition of a regeneration speed comparable to that of the Ag group, whose graft had the tracheal structures during the surgery. We consider the presence of these vessels to be one of the major advantages of the aTET when functioning as a tracheal graft. Regarding cartilage calcification, it exists in the autograft, as shown in Figure 6. So, we think that it does not affect strength and that it does not have a significant in the long term. Some reports have reported that calcification should be avoided [2,3,4,5,6], but it remains unclear how calcification affects the strength of a trachea.

Figure 7 shows a schema of the regeneration processes of the three grafts based on the histological findings observed in this study. In aTET grafts or autografts, tracheal regeneration progressed with a reduction in inflammatory response about 1 month after surgery, and complete regeneration was achieved within a few months. On the other hand, in the synthetic grafts, the inflammatory reaction was finally reduced several months after surgery. However, the inflammation around the synthetic grafts remained. Moreover, tracheal regeneration was incomplete, and no ciliary cells or goblet cells were observed.

Many of the benefits of an aTET are referred to here, but this study has two limitations. One limitation was that we performed patch tracheoplasty, not circumferential tracheal transplantation. It is not possible to conclude whether an aTET can truly solve the three aforementioned problems. In addition, whereas cells and growth factors can be expected to migrate from all directions around a graft in the case of patch tracheoplasty, they only migrate from the head or caudal direction in the case of circumferential tracheal replacement. So, it is unknown whether circumferential tracheal replacement has the same regenerative process. Differences in the number of sutures ranged from six to eight, but within this range, no findings in the present study indicate that a difference was made in terms of tissue regeneration. Thus, we speculate that the number does not affect regenerative processes, even in the case of the circumferential tracheal implantation. The other limitation was that we could not perform a histological evaluation in the same region in all groups. It would be ideal to perform a tissue evaluation at the same location in each group and present images. However, the changes occurring in each graft vary in location and degree, making it difficult to perform an evaluation at the same location. Note that macroscopically, all tissue evaluations were performed in the center of the graft.

Even with the mentioned limitations, the knowledge provided by our study will be very useful for research aimed at clinical applications. To date, in most related experiments, large animals were mainly used because clinical interventions of stent insertion and dilation or cauterization with bronchial fibers are available for maintaining luminal space after surgery. Strength problems can be managed with stents, granulation can be treated with tracheal dilatation or a granulation cautery under general anesthesia, and problems with sputum evacuation due to poor epithelial regeneration can be treated with endotracheal suctioning. For these reasons, we speculate that the animal models used for circumferential tracheal replacement were large animals since small animal models have cannot survive for the required observation period. From a viewpoint of clinical practice and research, this report will be able to serve as a trigger for future experiments on small animals.

## 5. Conclusions

We confirmed that the aTET reproducibly allowed complete tracheal regeneration of the tissue-engineered grafts within 3–6 months. Our data imply that an aTET has the potential to be a suitable tracheal graft in the case of circumferential tracheal replacement.

## Figures and Tables

**Figure 1 bioengineering-11-00243-f001:**
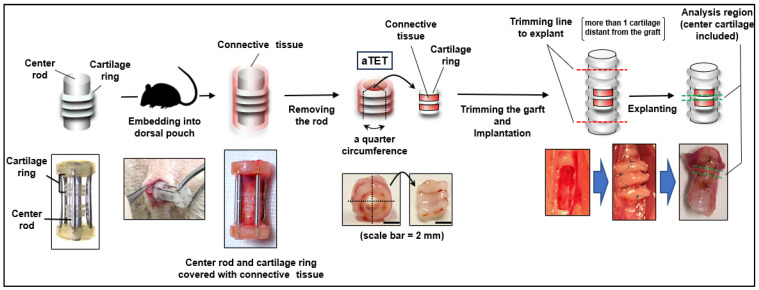
The procedure for autologous Tissue-Engineered Trachea (aTET) preparation and patch implantation, describing the extent of resection.

**Figure 2 bioengineering-11-00243-f002:**
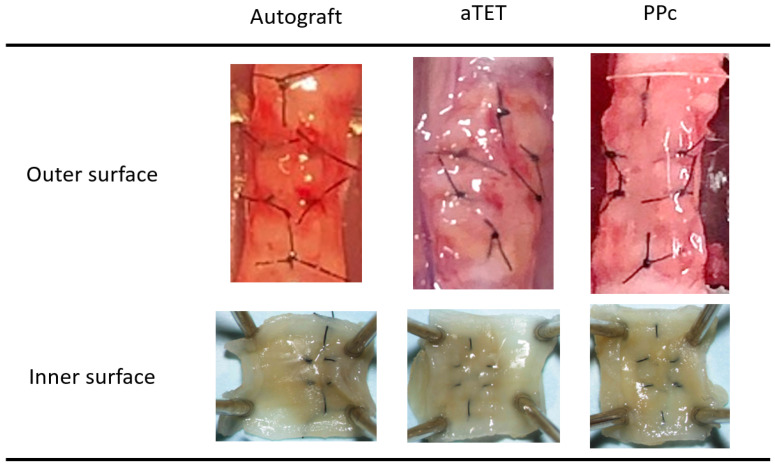
Gross appearance: outer and inner surfaces of explanted grafts after 1 month postoperatively (PPc: polylactic acid and polycaprolactone copolymer).

**Figure 7 bioengineering-11-00243-f007:**
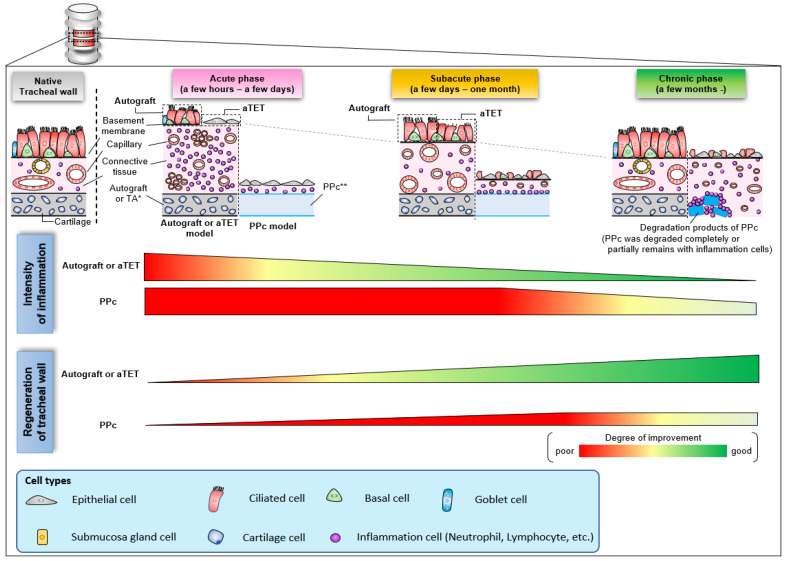
Schema of histological change with the trend of inflammation and regeneration in the intratracheal environment.

## Data Availability

The data are not publicly available due to their containing information that could compromise the privacy of research participants.

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
