# Peer review of "Regeneration Process of an Autologous Tissue-Engineered Trachea (aTET) in a Rat Patch Tracheoplasty Model"

_bioengineering, 2024, doi:10.3390/bioengineering11030243_

Round 1

Reviewer 1 Report

Comments and Suggestions for Authors

Overall comment: The authors have successfully created an artificial trachea tissue (aTET) using a combination of collagen and cartilage-like tissue. The manuscript focuses on in vivo studies that investigate the regenerative potential of aTET, showing promising results for its potential as an autograft replacement in the future. I have a few minor suggestions for refinement to improve the manuscript:

1.     Consider replacing the images in Figure 5 with higher resolution versions. Also, in Figure 5r, there seems to be an inconsistancy in magnitude, as the size of the nucleus appears different compared to other images.

2.     You may want to include an illustration in Figure 1 to visually depict the structural distribution of collagen and cartilage-like tissue within aTET.

3.     To improve clarity, it is recommended to integrate the subfigures in Figure 3 and Figure 4 and consolidate the corresponding figure legends.

4.     The rationale for selecting the dorsal pouch as the embedding site for aTET should be explicitly described.

5.     To enhance the visual representation of the survival rate, consider presenting it in the form of a graph.

6.     Immunostaining results should be quantified to enable a more rigorous comparison and clearer interpretation of the findings.

Comments on the Quality of English Language

Minor editing can be considered.

Author Response

Thank you very much for reviewing our manuscript and offering valuable advice.

We have addressed your comments with point-by-point responses, and revised the manuscript accordingly

  1. Consider replacing the images in Figure 5 with higher resolution versions. Also, in Figure 5r, there seems to be an inconsistancy in magnitude, as the size of the nucleus appears different compared to other images.

I revised Figure 5.

  1. You may want to include an illustration in Figure 1 to visually depict the structural distribution of collagen and cartilage-like tissue within aTET.

  I added the structural distribution of aTET in Figure 1.

  1. To improve clarity, it is recommended to integrate the subfigures in Figure 3 and Figure 4 and consolidate the corresponding figure legends.

  I integrated the subfigures in Figure 3 and 4, including each legends.

  1. The rationale for selecting the dorsal pouch as the embedding site for aTET should be explicitly described.

  The implantation in the dorsal pouch reduces the rat's body movement effect on the mold, which could avoid cartilage deviation to produce aTETs with an almost similar shape with reproducibility. 

  1. To enhance the visual representation of the survival rate, consider presenting it in the form of a graph.

  In this paper, our experiments were carried out on the assumption that the patch implantation has a high survival rate. Since the postoperative survival rate was 100%, excluding intraoperative deaths, we do not consider the graph of survival rates necessary.

  1. Immunostaining results should be quantified to enable a more rigorous comparison and clearer interpretation of the findings.

  Quantitative assessment of inflammation is shown in Figures 3B and 4B. No quantitative evaluation was made for Figure 5 because the results are clear.

Reviewer 2 Report

Comments and Suggestions for Authors

The manuscript entitled “Regeneration process of an autologous tissue engineered trachea (aTET) in a rat patch tracheoplasty model” shows an experimental work performed by a group of research which has been working on tissue engineering, particularly focused to trachea repair. Authors previously published a paper regards a construct that mimics a segment of a rat trachea capable to replace circumferential damages, but the efficiency of the construct/model was very low (60% survival). Apparently, with this manuscript, authors have considered to use the same construct, but focused to partial damage of the trachea, such as a patch. This is a good idea, because the limited damage and patch treatment has led them to improve significantly animal’s survival. Nevertheless, in the opinion of this reviewer, the previous work and the present, lack of appropriate histomorphological evaluation and take it for granted several considerations regards the apparent tissue “regeneration”. Besides, some methodological aspects should be revised in order to be considered for its publication in Bioengineering.

Following, you can find the issues that this reviewer considers should be addressed before manuscript acceptance:

Introduction section.

1. What is the aim of the Table, whether it indicates "Reports of circumferential tracheal reconstruction with long post-operative survive", and this manuscript refers to patch repair.

Materials and methods section.

2. The process evaluated in the previous work (ref. 32) referred a whole segment trachea reparation, meanwhile in this work authors modeled patch repair. Differences between whole section or trachea patch repair are quite different, since the authors declared in their previous work " The patch implantation of the Biosheets into the partial tracheal defects in beagle dogs resulted in the reconstruction of the tracheal tissue by covering the ciliated epithelium on the lumen surface and the partial infiltration of chondrocyte aggregates in the middle wall. When the Biotube was implanted circumferentially into the trachea, stenosis occurred within a few days owing to its insufficient strength.". For that reason, this reviewer suggests authors should revise the paragraph and the information they would like to introduce to the readers.

3. 2.1 Graft preparation section.

• As the Figure 1 includes dorsal pouch embedding, authors should mention this figure in this section.

• Line 84. Authors should describe or cite how PPc was handled. It means, PPc was placed into dorsal subcutaneous pouch also?

4. 2.3 Evaluations of the explanted graft section.

• This reviewer suggests that authors use the whole name for every staining.

• Line 128. How explant graft sections were obtained. This reviewer suggests authors prepare an image to be more explicit.

• Line 138. Why authors used alpha SMA as fibroblast marker, since the protein is characteristic of myofibroblasts and the corresponding undesirable fibrogenesis.

• Authors should include the fluorochromes used and whether the primary antibody was conjugated.

• Line 143. Authors mean "microscope Eclipse"?

Results section.

5. Since the work lacks of a diagram of the tissue sections obtained for histology, it is necessary that authors include it and show the regions for morphological analysis.

6. Authors should revise the section 3.1 Survival rate. In the line 160, authors declare "The overall postoperative survival rate was ...", meanwhile in the lines 168 and 169 they conclude: "Both of the deaths of the two rats were intraoperative deaths by the intraluminal sputum. No postoperative death was seen." If no postoperative deaths were seen, the overall postoperative survival is different.

7. Figure 2. According to the figure, the different grafts were fixed with different number of sutures; 7 for autografts, 8 for aTET and 6 for PPc. This, could be a model bias, since aTET group had 25% more suture effects than PPc. Authors should recognize it and discuss the relevance of every procedure.

8. Figure 2. What is the aim of asterisk(s)? In the opinion of this reviewer asterisks are unnecessary.

9. 3.2 Gross appearance of the graft section. What does "maintained well" means? Please clarify.

10. 3.3 Histological examinations. The section shown for Masson trichrome staining from PPc group did not show the surface of the graft, as it was observed for the other two groups. So, with the low magnification in photomicrograph 3 is very difficult to know if the epithelium was absent; indeed, there is a hematoxylin stained cell layer.

11. Why authors show sections from different regions, instead of to show the same region for every graft. This could be more appropriate, mainly if they are trying to demonstrate structural changes among groups.

12. Do the authors assessed group statistical differences on every time? It means, Autograft, aTET and PPc at the 1st month, and so forth.

13. Photomicrographs from IFs are overexposed, DAPI was too high, so it is difficult to evaluated fluorochrome expression. As the images have been obtained at high magnification, it is not possible to know the location of the expression of CD68 and alpha-SMA; usually it should be from the same region from each tissue.

14. Line 210 and others. Authors mean tubulin?

15. Figure 5. Why inner and outer regions were only shown for PPc group? It could be interesting to show for all the groups.

16. According to images, aTET exhibits positive Von Kossa staining also. Please revise it or discuss it. It could be desirable that authors integrate the information regards mineralization of trachea cartilage in this section, as well as the meaning of that process.

17. Figure 3A. Please state "PO" meaning.

18. Figure 3B and other charts. This reviewer suggests that authors use "y" axis line and improve the quality of every chart.

19. Figure 6. Please organize the figure in ascending manner, it means a-f.

Discussion.

20. This section depends completely of the results that should be revised. Authors’ statements are based on qualitative analyses, despite they have performed statistical assessments for the histomorphological changes. If there were no statistically significant difference, then the model is not enough to distinguish change among groups.

21. Please avoid to repeat results in this section.

22. Line 267. Authors mean “have been”?

23. Line 274. Please use italics for et al.

24. Line 296. Please avoid contractions.

25. Line 297. But, in this work trachea replacement was a patch. So, authors should clarify this sentence.

Conclusions section.

26. This section is not supported with the results shown in the manuscript.

Author Response

Thank you very much for reviewing our manuscript and offering valuable advice.

We have addressed your comments with point-by-point responses, and revised the manuscript accordingly.

Introduction section.

  1. What is the aim of the Table, whether it indicates "Reports of circumferential tracheal reconstruction with long post-operative survive", and this manuscript refers to patch repair.

The aim of the table is mainly to convey 3 points; 1) there are few reports on long-term survival models of circumferential tracheal replacement, 2) the reproducible protocol of the model was not demonstrated within each report, 3)there are especially few reports on small animal experimental models, which lacks versatility for basic research.

Materials and Methods section.

  1. The process evaluated in the previous work (ref. 32) referred a whole segment trachea reparation, meanwhile in this work authors modeled patch repair. Differences between whole section or trachea patch repair are quite different, since the authors declared in their previous work " The patch implantation of the Biosheets into the partial tracheal defects in beagle dogs resulted in the reconstruction of the tracheal tissue by covering the ciliated epithelium on the lumen surface and the partial infiltration of chondrocyte aggregates in the middle wall. When the Biotube was implanted circumferentially into the trachea, stenosis occurred within a few days owing to its insufficient strength.". For that reason, this reviewer suggests authors should revise the paragraph and the information they would like to introduce to the readers.

The purpose of this report is to introduce the reader to the regeneration process of grafts, which has never been shown before, using a patch grafting model that provides high survival rates even in small animals, and to evaluate it both grossly and histologically in various setting periods.

  1. Graft preparation section.
    • As the Figure 1 includes dorsal pouch embedding, authors should mention this figure in this section.

I added the picture of dorsal pouch embedding in Figure 1.

    • Line 84. Authors should describe or cite how PPc was handled. It means, PPc was placed into dorsal subcutaneous pouch also?

PPc was implanted directly into the rat trachea, not implanted into the dorsal pouch before patch tracheoplasty.

  1. 2.3 Evaluations of the explanted graft section.
    • This reviewer suggests that authors use the whole name for every staining.

All stained names have been rewritten to full names.

    • Line 128. How explant graft sections were obtained. This reviewer suggests authors prepare an image to be more explicit.

Trimming lines to explant the graft have been added in Figure 1.

    • Line 138. Why authors used alpha SMA as fibroblast marker, since the protein is characteristic of myofibroblasts and the corresponding undesirable fibrogenesis.

Immunostaining with anti-CD68 and anti-alpha SMA in this study was performed in the former for qualitative assessment of inflammatory intensity and in the latter for assessment of granulation by myofibroblasts following inflammation. The phrase "as a fibroblast marker" (Line 138-139) was strictly incorrect.

In the Autograft and aTET groups, alpha-SMA did not appear after the loss of CD68, whereas in the PPc group, CD68 remained, and alpha-SMA-positive cells were also abundant in the granulation, as shown in Fig. 4A.

The phrase " healing tendency" in the Discussion section may be misleading and has been reworded as follows.

“The use of aTET resulted in quiescence of inflammation without granulation, whereas the use of PPc was likely to cause persistent inflammation and granulation, which may be significant risk for airway structures with narrow lumens.”

    • Authors should include the fluorochromes used and whether the primary antibody was conjugated.

Secondary antibody information was omitted and has been added to the manuscript.

    • Line 143. Authors mean "microscope Eclipse"?

The description has been corrected as follows;

We used a fluorescence microscope (Eclipse-Ti, Nikon®), analysis software (NIS-Elements, Nikon®)...

Results section.

  1. Since the work lacks of a diagram of the tissue sections obtained for histology, it is necessary that authors include it and show the regions for morphological analysis.

The explanation was added in Figure 1.

  1. Authors should revise the section 3.1 Survival rate. In the line 160, authors declare "The overall postoperative survival rate was ...", meanwhile in the lines 168 and 169 they conclude: "Both of the deaths of the two rats were intraoperative deaths by the intraluminal sputum. No postoperative death was seen." If no postoperative deaths were seen, the overall postoperative survival is different.

The manuscript has been revised as noted by the reviewer.

  1. Figure 2. According to the figure, the different grafts were fixed with different number of sutures; 7 for autografts, 8 for aTET and 6 for PPc. This, could be a model bias, since aTET group had 25% more suture effects than PPc. Authors should recognize it and discuss the relevance of every procedure.

The number of sutures is adjusted during surgery as needed to maintain the shape and airtightness of the graft. In addition to this, although the number of ligatures in the same group sometimes differed, this did not make a difference in respiratory symptoms or the regenerative process.

  1. Figure 2. What is the aim of asterisk(s)? In the opinion of this reviewer asterisks are unnecessary.

The asterisks has been omitted as noted by the reviewer.

  1. 3.2 Gross appearance of the graft section. What does "maintained well" means? Please clarify

The phrase "maintained well" has been revised to a different expression and the manuscript has been rewritten.

  1. 3.3 Histological examinations. The section shown for Masson trichrome staining from PPc group did not show the surface of the graft, as it was observed for the other two groups. So, with the low magnification in photomicrograph 3 is very difficult to know if the epithelium was absent; indeed, there is a hematoxylin stained cell layer.

The process of epithelial regeneration was evaluated by Hematoxylin and Eosin staining, Alcian Blue staining, and immunofluorescence staining for alpha Tubulin. Masson’s Trichrome was performed to observe the presence of granulation tissue. In the PPc group with thickened submucosal tissue, the epithelium was not included in the same image accidentally.

  1. Why authors show sections from different regions, instead of to show the same region for every graft. This could be more appropriate, mainly if they are trying to demonstrate structural changes among groups.

There is cartilage tissue in the autograft and aTET groups and PPc itself in the PPc group, which limits the areas where inflammation, angiogenesis, and granulation can be evaluated histologically. Therefore, it is technically difficult to analyze almost the same areas in each group. Therefore, the images of the most suitable areas for observation in each group were presented for the characteristics of tissues other than cartilage and PPc.

  1. Do the authors assessed group statistical differences on every time? It means, Autograft, aTET and PPc at the 1st month, and so forth.

We want to show that inflammation, angiogenesis and granulation are decreased in the Autograft and aTET groups and increased in the PPc group. In other words, we do not want to show significant differences in quantitative evaluations between groups, but rather the characteristics of the changes in each graft over time.

  1. Photomicrographs from IFs are overexposed, DAPI was too high, so it is difficult to evaluated fluorochrome expression. As the images have been obtained at high magnification, it is not possible to know the location of the expression of CD68 and alpha-SMA; usually it should be from the same region from each tissue.

As much as possible, we adjusted the brightness and contrast so that each fluorochrome expression is easily visible.

About the magnification, the pictures was taken at the magnification presented because the area where inflammation, angiogenesis, and granulation can be histologically evaluated is limited as mentioned in response to comment #11 and lower magnification make it difficult to confirm CD68 which is present in minute amounts in the plasma membrane, as opposed to alpha SMA which is present in the cytoplasm. We have confirmed that this is the site of the overall feature.

  1. Line 210 and others. Authors mean tubulin?

It is alpha Tubulin. The manuscript has been corrected.

  1. Figure 5. Why inner and outer regions were only shown for PPc group? It could be interesting to show for all the groups.

The purpose of Figure 5 is to evaluate epithelial regeneration, which is important for survival-related respiratory function, and granulation of submucosal tissue, which can lead to luminal narrowing. The 3- and 6-month models of the PPc group had extremely thickened subcutaneous tissue, and the epithelium and subcutaneous tissue could not be included in the same image, so only the PPc group was shown in two separate images.

  1. According to images, aTET exhibits positive Von Kossa staining also. Please revise it or discuss it. It could be desirable that authors integrate the information regards mineralization of trachea cartilage in this section, as well as the meaning of that process.

In the aTET group, calcification was first observed at 6 months (Figure 6 (f)). Red circles surrounding areas of calcification have been added to Figure 6(f) for easier visualization.

  1. Figure 3A. Please state "PO" meaning.

PO means “postoperatively”, and the full spell has been added to the manuscript.

  1. Figure 3B and other charts. This reviewer suggests that authors use "y" axis line and improve the quality of every chart.

The Y-axis has been added and modified to make the graph easier to read.

  1. Figure 6. Please organize the figure in ascending manner, it means a-f.

In Figure 6, the images has been changed to ascending manner as noted by the reviewer.

Discussion

  1. This section depends completely of the results that should be revised. Authors’ statements are based on qualitative analyses, despite they have performed statistical assessments for the histomorphological changes. If there were no statistically significant difference, then the model is not enough to distinguish change among groups.

As I replied in comment #12, it is not the main purpose of this paper to show significant differences between groups. Statistical comparisons were performed in each group, and these were used to characterize the histology of the grafts themselves.

Suppose that the PPc is a potentially ideal graft that shows a decreasing trend in inflammation, angiogenesis, and granulation. In this case, we would need to statistically examine the differences between the groups to show whether aTET or PPc is the better graft, as the reviewer pointed out. However, since PPc was found to be associated with increased inflammation and granulation, it was presented only as a qualitative comparison.

Since PPc is a synthetic material that has been used clinically  and has been used in tracheal transplantation studies, it was used as a representative synthetic material in this study.

  1. Please avoid to repeat results in this section.

The manuscript was rewritten to avoid repeating results.

The specific locations that were deleted and modified are shown below (Lines in the Manuscript first sent); Lines 293-295, 300-307, 321-322, 325-333.

  1. Line 267. Authors mean “have been”?

 The manuscript has been revised as noted by the reviewer.

  1. Line 274. Please use italics for et al.

The manuscript has been revised as noted by the reviewer.

  1. Line 296. Please avoid contractions.

The manuscript has been revised as noted by the reviewer.

  1. Line 297. But, in this work trachea replacement was a patch. So, authors should clarify this sentence.

The description from line 297 is changed as follows;

If circumferential tracheal replacement was performed instead of patch implantation...

Conclusion section.

  1. This section is not supported with the results shown in the manuscript.

 As noted by the reviewer, the strength cannot be supported by the results of this study, so the sentence was changed as follows;

 We could confirm the complete tracheal regeneration process with reproducibility in the aTET of the tissue-engineered grafts within 3-6 months. Our data imply that aTET has a potential to be a suitable tracheal graft in the case of circumferential tracheal replacement.

Round 2

Reviewer 2 Report

Comments and Suggestions for Authors

Author Response

Dear Editors and Reviewers

Thank you very much for reviewing our manuscript and offering valuable advice again.

We have addressed your round 2 comments with a point-by-point response and revised the manuscript accordingly.

Note that, the line number in this response was assigned in the revised manuscript.

Introduction section.

  1. The original question was: What is the aim of the Table, whether it indicates "Reports of circumferential tracheal reconstruction with long post-operative survive", and this manuscript refers to patch repair?

Authors’ response was: The aim of the table is mainly to convey 3 points; 1) there are few reports on long-term survival models of circumferential tracheal replacement, 2) the reproducible protocol of the model was not demonstrated within each report, 3)there are especially few reports on small animal experimental models, which lacks versatility for basic research.

However, this manuscript refers to a patch model. In the opinion of this reviewer a short sentence should be enough to justify that there are few reports on long-term survival model of circumferential tracheal replacement; and then focus on their patch model.

If only to state the reason for performing patch tracheoplasty in this experiment, it is enough to add the short sentence as the reviewer commented for us.

When we submitted this paper, we wanted to address the fact that circumferential tracheal transplantation has been performed in situations where long survival is unlikely and that no previous reports have attempted to demonstrate the reproducibility of long survival. We also hoped that this paper would serve as a starting point to help demonstrate the reproducibility of the long survival. However, we have decided to omit Table 1 from this paper as it has little relevance to what we wish to discuss in this paper. We will consider presenting this table again in the future when we perform the experiment of circumferential tracheal transplantation.

  1. 1 Graft preparation section.
  • The original question was: Line 84. Authors should describe or cite how PPc was handled. It means, PPc was placed into dorsal subcutaneous pouch also?

Authors’ response was: PPc was implanted directly into the rat trachea, not implanted into the dorsal pouch before patch tracheoplasty.

Please include the sentence in the manuscript.

I have added the sentence in the Materials and Methods section (Line 103-104).

  1. 3 Evaluations of the explanted graft section.
  • The original question was: Line 138. Why authors used alpha SMA as fibroblast marker, since the protein is characteristic of myofibroblasts and the corresponding undesirable fibrogenesis.

Authors’ response was: Immunostaining with anti-CD68 and anti-alpha SMA in this study was performed in the former for qualitative assessment of inflammatory intensity and in the latter for assessment of granulation by myofibroblasts following inflammation. The phrase "as a fibroblast marker" (Line 138-139) was strictly incorrect.

Precisely, “as a fibroblast marker” is strictly incorrect. If authors recognize the information, why in the lines 138 and 139 still appears as “and anti-alpha SMA antibody (1:100, 138 904601, BioLegend) for fibroblast cells…”?

The following text was added in the section (Line 162-164);

When the granulation tissue is formed, the fibroblasts differentiate and the myofibroblasts appear, which are observed and evaluated qualitatively.

Results section.

  1. The original question was: Figure 2. According to the figure, the different grafts were fixed with different number of sutures; 7 for autografts, 8 for aTET and 6 for PPc. This, could be a model bias, since aTET group had 25% more suture effects than PPc. Authors should recognize it and discuss the relevance of every procedure.

Authors’ response was: The number of sutures is adjusted during surgery as needed to maintain the shape and airtightness of the graft. In addition to this, although the number of ligatures in the same group sometimes differed, this did not make a difference in respiratory symptoms or the regenerative process.

This reviewer disagrees regards to the concept of the number of sutures has not effect on the regeneration process. However, if the authors have a solid information about that, they should discuss it with supplementary material or a reference. And, in the event authors demonstrate the lack of adverse effects due to the number of sutures, they should include it in the Discussion section.

The following text was added in the Discussion section (Line 398-400);

Differences in the number of sutures ranged from 6 to 8. Within this range, no findings in the present study have made a difference in tissue regeneration.

  1. The original question was: 3 Histological examinations. The section shown for Masson

trichrome staining from PPc group did not show the surface of the graft, as it was observed for the other two groups. So, with the low magnification in photomicrograph 3 is very difficult to know if the epithelium was absent; indeed, there is a hematoxylin stained cell layer.

Authors’ response was: Masson’s Trichrome was performed to observe the presence of granulation tissue. In the PPc group with thickened submucosal tissue, the epithelium was not included in the same image accidentally.

Authors should include the appropriate image.

Images containing the epithelial layer are added in (l), (q), and (r) of FIgure 3A, where the epithelial layer was not included in the same picture. The location of the added image is indicated by the addition of a rectangle in the Hematoxylin and Eosin staining.

  1. The original question was: Why authors show sections from different regions, instead of to show the same region for every graft. This could be more appropriate, mainly if they are trying to demonstrate structural changes among groups.

Authors’ response was: There is cartilage tissue in the autograft and aTET groups and PPc itself in the PPc group, which limits the areas where inflammation, angiogenesis, and granulation can be evaluated histologically. Therefore, it is technically difficult to analyze almost the same areas in each group. Therefore, the images of the most suitable areas for observation in each group were presented for the characteristics of tissues other than cartilage and PPc.

However, the manuscript is based on the histological characteristics of the different therapeutic strategies. So, the appropriate images should be included.

As the reviewer pointed out, it would be ideal to perform a tissue evaluation at the same location in each group and present images. However, the changes occurring in each graft vary in location and degree, making it difficult to evaluate at the same location. This is the limitation of this study. Note that, macroscopically, all tissue evaluations were performed in the center of the graft.

This idea was added in the Discussion section(Line 404-406).

  1. The original question was: Do the authors assessed group statistical differences on every time? It means, Autograft, aTET and PPc at the 1st month, and so forth.

Authors’ response was: We want to show that inflammation, angiogenesis and granulation are decreased in the Autograft and aTET groups and increased in the PPc group. In other words, we do not want to show significant differences in quantitative evaluations between groups, but rather the characteristics of the changes in each graft over time.

First, authors had performed statistical analyses. So, they attempt to demonstrate a qualitative assessment. Secondly, this reviewer considers that comparison among the three times evaluated should be compared in order to demonstrate changes over time, that is precisely what the authors want.

It is certainly a small number of samples to make a statistical evaluation, and our emphasis on reproducibility may be an overstatement of our reproducibility statements. In addition, by stating, as we did in our initial reply to the reviewer's correction, that PPc tends to exacerbate inflammation and autograft and aTET tend to alleviate inflammation, we may be confusing the reader as to whether we are primarily addressing a qualitative or quantitative assessment of reproducibility in this paper. Therefore, given the difficulty of statistical evaluation in this paper and the possibility of confusing interpretations, we have omitted statistical evaluation in this paper and used only qualitative evaluation in this paper.

Accordingly, the section of “2.5 Statistical analysis” has been deleted, the reason why we omit it was added (Line 151-153) in “2.3 Evaluaions of the explanted graft”, and the comments on statistical evaluation in the Results and Discussion sections have been removed.

  1. The original question was: Photomicrographs from IFs are overexposed, DAPI was too high, so it is difficult to evaluated fluorochrome expression. As the images have been obtained at high magnification, it is not possible to know the location of the expression of CD68 and alpha-SMA; usually it should be from the same region from each tissue.

Authors’ response was: About the magnification, the pictures was taken at the magnification presented because the area where inflammation, angiogenesis, and granulation can be histologically evaluated is limited as mentioned in response to comment #11 and lower magnification make it difficult to confirm CD68 which is present in minute amounts in the plasma membrane, as opposed to alpha SMA which is present in the cytoplasm. We have confirmed that this is the site of the overall feature.

This reviewer understands the authors’ intention, but precisely because the different localization between CD68 and αSMA positive cells, it would be desirable to show low magnification photomicrographs. In the form that the IF images have been included in the manuscript they show nothing.

Immunofluorescent staining image is generally difficult to recognize in low-magnification. Therefore, we did not save low magnification images of immunofluorescence staining for the alpha SMA and the CD68. However, because the tissues formed in the autograft and aTET groups were thin, we could observe the full thickness of these graft walls even at the original magnification. In addition, the distribution of alpha SMA- and CD68-positive cells in the grafts was almost the same in the autograft and aTET groups.

On the other hand, as for the 3- and 6-month models in the PPc group, as shown in Figures 3A and 5, we left images for both of the outer and inner areas because of the strong thickening of the submucosal tissue. Therefore, in Figure 4A, we presented two images of the inner and outer areas, where the presence of cell types differed, only for the 3- and 6-month models of the PPc group.

The contents of the results (Line 224-228) and discussion sections (Line 338-339) were also revised accordingly.

  1. The original question was: Figure 5. Why inner and outer regions were only shown for PPc group? It could be interesting to show for all the groups.

Authors’ response was: The purpose of Figure 5 is to evaluate epithelial regeneration, which is important for survival-related respiratory function, and granulation of submucosal tissue, which can lead to luminal narrowing. The 3- and 6-month models of the PPc group had extremely thickened subcutaneous tissue, and the epithelium and subcutaneous tissue could not be included in the same image, so only the PPc group was shown in two separate images.

So, please include the idea in the Results and/or Discussion sections.

I inserted this idea in the Results section (Line 232-234).

  1. The original question was: According to images, aTET exhibits positive Von Kossa staining also. Please revise it or discuss it. It could be desirable that authors integrate the

information regards mineralization of trachea cartilage in this section, as well as the meaning of that process.

Authors’ response was: In the aTET group, calcification was first observed at 6 months (Figure 6 (f)). Red circles surrounding areas of calcification have been added to Figure 6(f) for easier visualization.

So, please include the idea in the Results and/or Discussion sections.

I inserted this idea in the Results section (Line 248-249).

Discussion.

  1. The original question was: This section depends completely of the results that should be revised. Authors’ statements are based on qualitative analyses, despite they have performed statistical assessments for the histomorphological changes. If there were no statistically significant difference, then the model is not enough to distinguish change among groups.

Authors’ response was:

As I replied in comment #12, it is not the main purpose of this paper to show significant differences between groups. Statistical comparisons were performed in each group, and these were used to characterize the histology of the grafts themselves.

Suppose that the PPc is a potentially ideal graft that shows a decreasing trend in inflammation, angiogenesis, and granulation. In this case, we would need to statistically examine the differences between the groups to show whether aTET or PPc is the better graft, as the reviewer pointed out. However, since PPc was found to be associated with increased inflammation and granulation, it was presented only as a qualitative comparison.

Since PPc is a synthetic material that has been used clinically and has been used in tracheal transplantation studies, it was used as a representative synthetic material in this study.

However, authors had performed statistical assessments, why those analyses should not be considered?

As I replied in #7, we have decided to omit it in this paper because the number of samples was too small for statistical evaluation, which may lead to overestimation of its interpretation and induce misinterpretation to the reader.
